# LEARNING DEEP REPRESENTATIONS BY MUTUAL INFORMATION ESTIMATION AND MAXIMIZATION

**R Devon Hjelm**
MSR Montreal, MILA, UdeM, IVADO
`devon.hjelm@microsoft.com`

**Alex Fedorov**
MRN, UNM

**Samuel Lavoie-Marchildon**
MILA, UdeM

**Karan Grewal**
U Toronto

**Phil Bachman**
MSR Montreal

**Adam Trischler**
MSR Montreal

**Yoshua Bengio**
MILA, UdeM, IVADO, CIFAR

## ABSTRACT

This work investigates unsupervised learning of representations by maximizing mutual information between an input and the output of a deep neural network encoder. Importantly, we show that structure matters: incorporating knowledge about locality in the input into the objective can significantly improve a representation's suitability for downstream tasks. We further control characteristics of the representation by matching to a prior distribution adversarially. Our method, which we call Deep InfoMax (DIM), outperforms a number of popular unsupervised learning methods and compares favorably with fully-supervised learning on several classification tasks in with some standard architectures. DIM opens new avenues for unsupervised learning of representations and is an important step towards flexible formulations of representation learning objectives for specific end-goals.

## 1 INTRODUCTION

One core objective of deep learning is to discover useful representations, and the simple idea explored here is to train a representation-learning function, i.e. an encoder, to maximize the mutual information (MI) between its inputs and outputs. MI is notoriously difficult to compute, particularly in continuous and high-dimensional settings. Fortunately, recent advances enable effective computation of MI between high dimensional input/output pairs of deep neural networks (Belghazi et al., 2018). We leverage MI estimation for representation learning and show that, depending on the downstream task, maximizing MI between the complete input and the encoder output (i.e., *global* MI) is often insufficient for learning useful representations. Rather, *structure matters*: maximizing the average MI between the representation and *local* regions of the input (e.g. patches rather than the complete image) can greatly improve the representation's quality for, e.g., classification tasks, while global MI plays a stronger role in the ability to reconstruct the full input given the representation.

Usefulness of a representation is not just a matter of information content: representational characteristics like independence also play an important role (Gretton et al., 2012; Hyvärinen & Oja, 2000; Hinton, 2002; Schmidhuber, 1992; Bengio et al., 2013; Thomas et al., 2017). We combine MI maximization with prior matching in a manner similar to adversarial autoencoders (AAE, Makhzani et al., 2015) to constrain representations according to desired statistical properties. This approach is closely related to the infomax optimization principle (Linsker, 1988; Bell & Sejnowski, 1995), so we call our method *Deep InfoMax* (DIM). Our main contributions are the following:

- We formalize Deep InfoMax (DIM), which simultaneously estimates and maximizes the mutual information between input data and learned high-level representations.

- Our mutual information maximization procedure can prioritize global or local information, which we show can be used to tune the suitability of learned representations for classification or reconstruction-style tasks.

- We use adversarial learning (à la Makhzani et al., 2015) to constrain the representation to have desired statistical characteristics specific to a prior.

- We introduce two new measures of representation quality, one based on Mutual Information Neural Estimation (MINE, Belghazi et al., 2018) and a neural dependency measure (NDM) based on the work by Brakel & Bengio (2017), and we use these to bolster our comparison of DIM to different unsupervised methods.

## 2  RELATED WORK

There are many popular methods for learning representations. Classic methods, such as independent component analysis (ICA, Bell & Sejnowski, 1995) and self-organizing maps (Kohonen, 1998), generally lack the representational capacity of deep neural networks. More recent approaches include deep volume-preserving maps (Dinh et al., 2014; 2016), deep clustering (Xie et al., 2016; Chang et al., 2017), noise as targets (NAT, Bojanowski & Joulin, 2017), and self-supervised or co-learning (Doersch & Zisserman, 2017; Dosovitskiy et al., 2016; Sajjadi et al., 2016).

Generative models are also commonly used for building representations (Vincent et al., 2010; Kingma et al., 2014; Salimans et al., 2016; Rezende et al., 2016; Donahue et al., 2016), and mutual information (MI) plays an important role in the quality of the representations they learn. In generative models that rely on reconstruction (e.g., denoising, variational, and adversarial autoencoders, Vincent et al., 2008; Rifai et al., 2012; Kingma & Welling, 2013; Makhzani et al., 2015), the reconstruction error can be related to the MI as follows:

$$\mathcal{I}_e(X, Y) = \mathcal{H}_e(X) - \mathcal{H}_e(X|Y) \geq \mathcal{H}_e(X) - \mathcal{R}_{e,d}(X|Y), \tag{1}$$

where $X$ and $Y$ denote the input and output of an encoder which is applied to inputs sampled from some source distribution. $\mathcal{R}_{e,d}(X|Y)$ denotes the expected reconstruction error of $X$ given the codes $Y$. $\mathcal{H}_e(X)$ and $\mathcal{H}_e(X|Y)$ denote the marginal and conditional entropy of $X$ in the distribution formed by applying the encoder to inputs sampled from the source distribution. Thus, in typical settings, models with reconstruction-type objectives provide some guarantees on the amount of information encoded in their intermediate representations. Similar guarantees exist for bi-directional adversarial models (Dumoulin et al., 2016; Donahue et al., 2016), which adversarially train an encoder / decoder to match their respective joint distributions or to minimize the reconstruction error (Chen et al., 2016).

**Mutual-information estimation**  Methods based on mutual information have a long history in unsupervised feature learning. The infomax principle (Linsker, 1988; Bell & Sejnowski, 1995), as prescribed for neural networks, advocates maximizing MI between the input and output. This is the basis of numerous ICA algorithms, which can be nonlinear (Hyvärinen & Pajunen, 1999; Almeida, 2003) but are often hard to adapt for use with deep networks. Mutual Information Neural Estimation (MINE, Belghazi et al., 2018) learns an estimate of the MI of continuous variables, is strongly consistent, and can be used to learn better implicit bi-directional generative models. Deep InfoMax (DIM) follows MINE in this regard, though we find that the generator is unnecessary. We also find it unnecessary to use the exact KL-based formulation of MI. For example, a simple alternative based on the Jensen-Shannon divergence (JSD) is more stable and provides better results. We will show that DIM can work with various MI estimators. Most significantly, DIM can leverage local structure in the input to improve the suitability of representations for classification.

Leveraging known structure in the input when designing objectives based on MI maximization is nothing new (Becker, 1992; 1996; Wiskott & Sejnowski, 2002), and some very recent works also follow this intuition. It has been shown in the case of discrete MI that data augmentations and other transformations can be used to avoid degenerate solutions (Hu et al., 2017). Unsupervised clustering and segmentation is attainable by maximizing the MI between images associated by transforms or spatial proximity (Ji et al., 2018). Our work investigates the suitability of representations learned across two different MI objectives that focus on local or global structure, a flexibility we believe is necessary for training representations intended for different applications.

Proposed independently of DIM, Contrastive Predictive Coding (CPC, Oord et al., 2018) is a MI-based approach that, like DIM, maximizes MI between global and local representation pairs. CPC shares some motivations and computations with DIM, but there are important ways in which CPC and DIM differ. CPC processes local features sequentially to build partial "summary features", which are used to make predictions about specific local features in the "future" of each summary feature. This equates to *ordered autoregression* over the local features, and requires training *separate estimators*

for each temporal offset at which one would like to predict the future. In contrast, the basic version of DIM uses a single summary feature that is a function of *all* local features, and this "global" feature predicts all local features simultaneously in a single step using *a single estimator*. Note that, when using occlusions during training (see Section 4.3 for details), DIM performs both "self" predictions and *orderless autoregression*.

## 3 DEEP INFOMAX

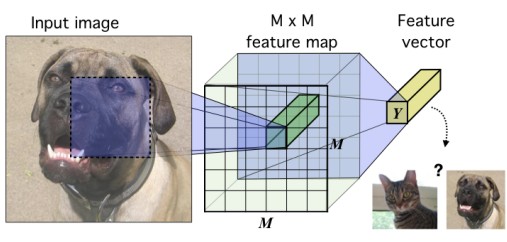

Figure 1: **The base encoder model in the context of image data.** An image (in this case) is encoded using a convnet until reaching a feature map of $M \times M$ feature vectors corresponding to $M \times M$ input patches. These vectors are summarized into a single feature vector, $Y$. Our goal is to train this network such that useful information about the input is easily extracted from the high-level features.

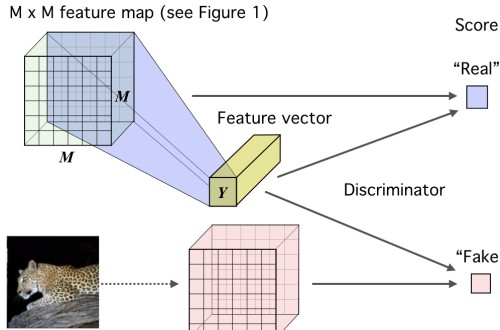

Figure 2: **Deep InfoMax (DIM) with a global MI$(X; Y)$ objective.** Here, we pass both the high-level feature vector, $Y$, and the lower-level $M \times M$ feature map (see Figure 1) through a discriminator to get the score. Fake samples are drawn by combining the same feature vector with a $M \times M$ feature map from another image.

Here we outline the general setting of training an encoder to maximize mutual information between its input and output. Let $\mathcal{X}$ and $\mathcal{Y}$ be the domain and range of a continuous and (almost everywhere) differentiable parametric function, $E_\psi : \mathcal{X} \to \mathcal{Y}$ with parameters $\psi$ (e.g., a neural network). These parameters define a family of encoders, $\mathcal{E}_\Phi = \{E_\psi\}_{\psi \in \Psi}$ over $\Psi$. Assume that we are given a set of training examples on an input space, $\mathcal{X}$: $\mathbf{X} := \{x^{(i)} \in \mathcal{X}\}_{i=1}^N$, with empirical probability distribution $\mathbb{P}$. We define $\mathbb{U}_{\psi, \mathbb{P}}$ to be the marginal distribution induced by pushing samples from $\mathbb{P}$ through $E_\psi$. I.e., $\mathbb{U}_{\psi, \mathbb{P}}$ is the distribution over encodings $y \in \mathcal{Y}$ produced by sampling observations $x \sim \mathcal{X}$ and then sampling $y \sim E_\psi(x)$.

An example encoder for image data is given in Figure 1, which will be used in the following sections, but this approach can easily be adapted for temporal data. Similar to the infomax optimization principle (Linsker, 1988), we assert our encoder should be trained according to the following criteria:

- **Mutual information maximization:** Find the set of parameters, $\psi$, such that the mutual information, $\mathcal{I}(X; E_\psi(X))$, is maximized. Depending on the end-goal, this maximization can be done over the complete input, $X$, or some *structured* or "local" subset.

- **Statistical constraints:** Depending on the end-goal for the representation, the marginal $\mathbb{U}_{\psi, \mathbb{P}}$ should match a prior distribution, $\mathbb{V}$. Roughly speaking, this can be used to encourage the output of the encoder to have desired characteristics (e.g., independence).

The formulation of these two objectives covered below we call *Deep InfoMax* (DIM).

### 3.1 MUTUAL INFORMATION ESTIMATION AND MAXIMIZATION

Our basic mutual information maximization framework is presented in Figure 2. The approach follows Mutual Information Neural Estimation (MINE, Belghazi et al., 2018), which estimates mutual information by training a classifier to distinguish between samples coming from the joint, $\mathbb{J}$, and the

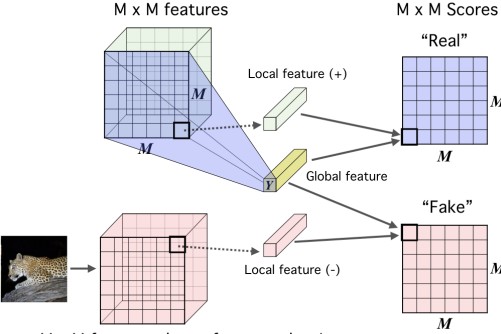

Figure 3: **Maximizing mutual information between local features and global features.** First we encode the image to a feature map that reflects some structural aspect of the data, e.g. spatial locality, and we further summarize this feature map into a global feature vector (see Figure 1). We then concatenate this feature vector with the lower-level feature map *at every location*. A score is produced for each local-global pair through an additional function (see the Appendix A.2 for details).

product of marginals, $\mathbb{M}$, of random variables $X$ and $Y$. MINE uses a lower-bound to the MI based on the Donsker-Varadhan representation (DV, Donsker & Varadhan, 1983) of the KL-divergence,

$$\mathcal{I}(X;Y) := \mathcal{D}_{KL}(\mathbb{J}||\mathbb{M}) \geq \widehat{\mathcal{I}}_\omega^{(DV)}(X;Y) := \mathbb{E}_\mathbb{J}[T_\omega(x,y)] - \log \mathbb{E}_\mathbb{M}[e^{T_\omega(x,y)}], \quad (2)$$

where $T_\omega : \mathcal{X} \times \mathcal{Y} \to \mathbb{R}$ is a discriminator function modeled by a neural network with parameters $\omega$.

At a high level, we optimize $E_\psi$ by simultaneously estimating and maximizing $\mathcal{I}(X, E_\psi(X))$,

$$(\hat{\omega}, \hat{\psi})_G = \arg\max_{\omega,\psi} \widehat{\mathcal{I}}_\omega(X; E_\psi(X)), \quad (3)$$

where the subscript $G$ denotes "global" for reasons that will be clear later. However, there are some important differences that distinguish our approach from MINE. First, because the encoder and mutual information estimator are optimizing the same objective and require similar computations, we share layers between these functions, so that $E_\psi = f_\psi \circ C_\psi$ and $T_{\psi,\omega} = D_\omega \circ g \circ (C_\psi, E_\psi)$,[1] where $g$ is a function that combines the encoder output with the lower layer.

Second, as we are primarily interested in maximizing MI, and not concerned with its precise value, we can rely on non-KL divergences which may offer favourable trade-offs. For example, one could define a Jensen-Shannon MI estimator (following the formulation of Nowozin et al., 2016),

$$\widehat{\mathcal{I}}_{\omega,\psi}^{(\text{JSD})}(X; E_\psi(X)) := \mathbb{E}_\mathbb{P}[-\text{sp}(-T_{\psi,\omega}(x, E_\psi(x)))] - \mathbb{E}_{\mathbb{P}\times\tilde{\mathbb{P}}}[\text{sp}(T_{\psi,\omega}(x', E_\psi(x)))], \quad (4)$$

where $x$ is an input sample, $x'$ is an input sampled from $\tilde{\mathbb{P}} = \mathbb{P}$, and $\text{sp}(z) = \log(1+e^z)$ is the softplus function. A similar estimator appeared in Brakel & Bengio (2017) in the context of minimizing the total correlation, and it amounts to the familiar binary cross-entropy. This is well-understood in terms of neural network optimization and we find works better in practice (e.g., is more stable) than the DV-based objective (e.g., see App. A.3). Intuitively, the Jensen-Shannon-based estimator should behave similarly to the DV-based estimator in Eq. 2, since both act like classifiers whose objectives maximize the expected log-ratio of the joint over the product of marginals. We show in App. A.1 the relationship between the JSD estimator and the formal definition of mutual information.

Noise-Contrastive Estimation (NCE, Gutmann & Hyvärinen, 2010; 2012) was first used as a bound on MI in Oord et al. (and called "infoNCE", 2018), and this loss can also be used with DIM by maximizing:

$$\widehat{\mathcal{I}}_{\omega,\psi}^{(\text{infoNCE})}(X; E_\psi(X)) := \mathbb{E}_\mathbb{P}\left[T_{\psi,\omega}(x, E_\psi(x)) - \mathbb{E}_{\tilde{\mathbb{P}}}\left[\log \sum_{x'} e^{T_{\psi,\omega}(x', E_\psi(x))}\right]\right]. \quad (5)$$

For DIM, a key difference between the DV, JSD, and infoNCE formulations is whether an expectation over $\mathbb{P}/\tilde{\mathbb{P}}$ appears inside or outside of a $\log$. In fact, the JSD-based objective mirrors the original NCE formulation in Gutmann & Hyvärinen (2010), which phrased unnormalized density estimation as binary classification between the data distribution and a noise distribution. DIM sets the noise distribution to the product of marginals over $X/Y$, and the data distribution to the true joint. The infoNCE formulation in Eq. 5 follows a softmax-based version of NCE (Jozefowicz et al., 2016), similar to ones used in the language modeling community (Mnih & Kavukcuoglu, 2013; Mikolov et al.,

---

[1]Here we slightly abuse the notation and use $\psi$ for both parts of $E_\psi$.

2013), and which has strong connections to the binary cross-entropy in the context of noise-contrastive learning (Ma & Collins, 2018). In practice, implementations of these estimators appear quite similar and can reuse most of the same code. We investigate JSD and infoNCE in our experiments, and find that using infoNCE often outperforms JSD on downstream tasks, though this effect diminishes with more challenging data. However, as we show in the App. (A.3), infoNCE and DV require a large number of *negative samples* (samples from $\tilde{\mathbb{P}}$) to be competitive. We generate negative samples using all combinations of global and local features at all locations of the relevant feature map, across all images in a batch. For a batch of size $B$, that gives $O(B \times M^2)$ negative samples per positive example, which quickly becomes cumbersome with increasing batch size. We found that DIM with the JSD loss is insensitive to the number of negative samples, and in fact outperforms infoNCE as the number of negative samples becomes smaller.

## 3.2 LOCAL MUTUAL INFORMATION MAXIMIZATION

The objective in Eq. 3 can be used to maximize MI between input and output, but ultimately this may be undesirable depending on the task. For example, trivial pixel-level noise is useless for image classification, so a representation may not benefit from encoding this information (e.g., in zero-shot learning, transfer learning, etc.). In order to obtain a representation more suitable for classification, we can instead maximize the average MI between the high-level representation and local patches of the image. Because the same representation is encouraged to have high MI with all the patches, this favours encoding aspects of the data that are shared across patches.

Suppose the feature vector is of limited capacity (number of units and range) and assume the encoder does not support infinite output configurations. For maximizing the MI between the whole input and the representation, the encoder can pick and choose what type of information in the input is passed through the encoder, such as noise specific to local patches or pixels. However, if the encoder passes information specific to only some parts of the input, this *does not increase* the MI with any of the other patches that do not contain said noise. This encourages the encoder to prefer information that is *shared* across the input, and this hypothesis is supported in our experiments below.

Our local DIM framework is presented in Figure 3. First we encode the input to a feature map, $C_\psi(x) := \{C_\psi^{(i)}\}_{i=1}^{M \times M}$ that reflects useful structure in the data (e.g., spatial locality), indexed in this case by $i$. Next, we summarize this local feature map into a global feature, $E_\psi(x) = f_\psi \circ C_\psi(x)$. We then define our MI estimator on global/local pairs, maximizing the average estimated MI:

$$(\hat{\omega}, \hat{\psi})_L = \arg\max_{\omega, \psi} \frac{1}{M^2} \sum_{i=1}^{M^2} \widehat{\mathcal{I}}_{\omega, \psi}(C_\psi^{(i)}(X); E_\psi(X)). \tag{6}$$

We found success optimizing this "local" objective with multiple easy-to-implement architectures, and further implementation details are provided in the App. (A.2).

## 3.3 MATCHING REPRESENTATIONS TO A PRIOR DISTRIBUTION

Absolute magnitude of information is only one desirable property of a representation; depending on the application, good representations can be compact (Gretton et al., 2012), independent (Hyvärinen & Oja, 2000; Hinton, 2002; Dinh et al., 2014; Brakel & Bengio, 2017), disentangled (Schmidhuber, 1992; Rifai et al., 2012; Bengio et al., 2013; Chen et al., 2018; Gonzalez-Garcia et al., 2018), or independently controllable (Thomas et al., 2017). DIM imposes statistical constraints onto learned representations by implicitly training the encoder so that the push-forward distribution, $\mathbb{U}_{\psi, \mathbb{P}}$, matches a prior, $\mathbb{V}$. This is done (see Figure 7 in the App. A.2) by training a discriminator, $D_\phi : \mathcal{Y} \to \mathbb{R}$, to estimate the divergence, $\mathcal{D}(\mathbb{V}||\mathbb{U}_{\psi, \mathbb{P}})$, then training the encoder to minimize this estimate:

$$(\hat{\omega}, \hat{\psi})_P = \arg\min_{\psi} \arg\max_{\phi} \widehat{\mathcal{D}}_\phi(\mathbb{V}||\mathbb{U}_{\psi, \mathbb{P}}) = \mathbb{E}_{\mathbb{V}}[\log D_\phi(y)] + \mathbb{E}_{\mathbb{P}}[\log(1 - D_\phi(E_\psi(x)))]. \tag{7}$$

This approach is similar to what is done in adversarial autoencoders (AAE, Makhzani et al., 2015), but without a generator. It is also similar to noise as targets (Bojanowski & Joulin, 2017), but trains the encoder to match the noise implicitly rather than using *a priori* noise samples as targets.

All three objectives – global and local MI maximization and prior matching – can be used together, and doing so we arrive at our complete objective for Deep InfoMax (DIM):

$$\arg\max_{\omega_1,\omega_2,\psi} \left( \alpha \widehat{\mathcal{I}}_{\omega_1,\psi}(X; E_\psi(X)) + \frac{\beta}{M^2} \sum_{i=1}^{M^2} \widehat{\mathcal{I}}_{\omega_2,\psi}(X^{(i)}; E_\psi(X)) \right) + \arg\min_{\psi} \arg\max_{\phi} \gamma \widehat{\mathcal{D}}_\phi(\mathbb{V}||\mathbb{U}_{\psi,\mathbb{P}}),$$
(8)

where $\omega_1$ and $\omega_2$ are the discriminator parameters for the global and local objectives, respectively, and $\alpha$, $\beta$, and $\gamma$ are hyperparameters. We will show below that choices in these hyperparameters affect the learned representations in meaningful ways. As an interesting aside, we also show in the App. (A.8) that this prior matching can be used alone to train a generator of image data.

## 4 EXPERIMENTS

We test Deep InfoMax (DIM) on four imaging datasets to evaluate its representational properties:

- CIFAR10 and CIFAR100 (Krizhevsky & Hinton, 2009): two small-scale labeled datasets composed of $32 \times 32$ images with 10 and 100 classes respectively.
- Tiny ImageNet: A reduced version of ImageNet (Krizhevsky & Hinton, 2009) images scaled down to $64 \times 64$ with a total of 200 classes.
- STL-10 (Coates et al., 2011): a dataset derived from ImageNet composed of $96 \times 96$ images with a mixture of 100000 unlabeled training examples and 500 labeled examples per class. We use *data augmentation* with this dataset, taking random $64 \times 64$ crops and flipping horizontally during unsupervised learning.
- CelebA (Yang et al., 2015, Appendix A.5 only): An image dataset composed of faces labeled with 40 binary attributes. This dataset evaluates DIM's ability to capture information that is more fine-grained than the class label and coarser than individual pixels.

For our experiments, we compare DIM against various unsupervised methods: Variational AutoEncoders (VAE, Kingma & Welling, 2013), $\beta$-VAE (Higgins et al., 2016; Alemi et al., 2016), Adversarial AutoEncoders (AAE, Makhzani et al., 2015), BiGAN (a.k.a. adversarially learned inference with a deterministic encoder: Donahue et al., 2016; Dumoulin et al., 2016), Noise As Targets (NAT, Bojanowski & Joulin, 2017), and Contrastive Predictive Coding (CPC, Oord et al., 2018). Note that we take CPC to mean ordered autoregression using summary features to predict "future" local features, independent of the constrastive loss used to evaluate the predictions (JSD, infoNCE, or DV). See the App. (A.2) for details of the neural net architectures used in the experiments.

### 4.1 HOW DO WE EVALUATE THE QUALITY OF A REPRESENTATION?

Evaluation of representations is case-driven and relies on various proxies. Linear separability is commonly used as a proxy for disentanglement and mutual information (MI) between representations and class labels. Unfortunately, this will not show whether the representation has high MI with the class labels when the representation is not disentangled. Other works (Bojanowski & Joulin, 2017) have looked at *transfer learning* classification tasks by freezing the weights of the encoder and training a small fully-connected neural network classifier using the representation as input. Others still have more directly measured the MI between the labels and the representation (Rifai et al., 2012; Chen et al., 2018), which can also reveal the representation's degree of entanglement.

Class labels have limited use in evaluating representations, as we are often interested in information encoded in the representation that is unknown to us. However, we can use mutual information neural estimation (MINE, Belghazi et al., 2018) to more directly measure the MI between the input and output of the encoder.

Next, we can directly measure the independence of the representation using a discriminator. Given a batch of representations, we generate a factor-wise independent distribution with the same per-factor marginals by randomly shuffling each factor along the batch dimension. A similar trick has been used for learning maximally independent representations for sequential data (Brakel & Bengio, 2017). We can train a discriminator to estimate the KL-divergence between the original representations (joint

distribution of the factors) and the shuffled representations (product of the marginals, see Figure 12). The higher the KL divergence, the more dependent the factors. We call this evaluation method *Neural Dependency Measure* (NDM) and show that it is sensible and empirically consistent in the App. (A.6).

To summarize, we use the following metrics for evaluating representations. For each of these, the encoder is held fixed unless noted otherwise:

- **Linear classification** using a support vector machine (SVM). This is simultaneously a proxy for MI of the representation with linear separability.

- **Non-linear classification** using a single hidden layer neural network (200 units) with dropout. This is a proxy on MI of the representation with the labels separate from linear separability as measured with the SVM above.

- **Semi-supervised learning** (STL-10 here), that is, fine-tuning the complete encoder by adding a small neural network on top of the last convolutional layer (matching architectures with a standard fully-supervised classifier).

- **MS-SSIM** (Wang et al., 2003), using a decoder trained on the $L_2$ reconstruction loss. This is a proxy for the total MI between the input and the representation and can indicate the amount of encoded pixel-level information.

- **Mutual information neural estimate (MINE)**, $\widehat{I}_\rho(X, E_\psi(x))$, between the input, $X$, and the output representation, $E_\psi(x)$, by training a discriminator with parameters $\rho$ to maximize the DV estimator of the KL-divergence.

- **Neural dependency measure (NDM)** using a second discriminator that measures the KL between $E_\psi(x)$ and a batch-wise shuffled version of $E_\psi(x)$.

For the neural network classification evaluation above, we performed experiments on all datasets except CelebA, while for other measures we only looked at CIFAR10. For all classification tasks, we built separate classifiers on the high-level vector representation ($Y$), the output of the previous fully-connected layer (fc) and the last convolutional layer (conv). Model selection for the classifiers was done by averaging the last 100 epochs of optimization, and the dropout rate and decaying learning rate schedule was set uniformly to alleviate over-fitting on the test set across all models.

## 4.2    REPRESENTATION LEARNING COMPARISON ACROSS MODELS

In the following experiments, DIM(G) refers to DIM with a global-only objective ($\alpha = 1, \beta = 0, \gamma = 1$) and DIM(L) refers to DIM with a local-only objective ($\alpha = 0, \beta = 1, \gamma = 0.1$), the latter chosen from the results of an ablation study presented in the App. (A.5). For the prior, we chose a compact uniform distribution on $[0, 1]^{64}$, which worked better in practice than other priors, such as Gaussian, unit ball, or unit sphere.

**Classification comparisons**    Our classification results can be found in Tables 1, 2, and 3. In general, DIM with the local objective, DIM(L), outperformed all models presented here by a significant margin on all datasets, regardless of which layer the representation was drawn from, with exception to CPC. For the specific settings presented (architectures, no data augmentation for datasets except for STL-10), DIM(L) performs as well as or outperforms a fully-supervised classifier without fine-tuning, which indicates that the representations are nearly as good as or better than the raw pixels given the model constraints in this setting. Note, however, that a fully supervised classifier can perform much better on all of these benchmarks, especially when specialized architectures and carefully-chosen data augmentations are used. Competitive or better results on CIFAR10 also exist (albeit in different settings, e.g., Coates et al., 2011; Dosovitskiy et al., 2016), but to our knowledge our STL-10 results are state-of-the-art for unsupervised learning. The results in this setting support the hypothesis that our local DIM objective is suitable for extracting class information.

Our results show that infoNCE tends to perform best, but differences between infoNCE and JSD diminish with larger datasets. DV can compete with JSD with smaller datasets, but DV performs much worse with larger datasets.

For CPC, we were only able to achieve marginally better performance than BiGAN with the settings above. However, when we adopted the strided crop architecture found in Oord et al. (2018), both

Table 1: Classification accuracy (top 1) results on CIFAR10 and CIFAR100. DIM(L) (i.e., with the local-only objective) outperforms all other unsupervised methods presented by a wide margin. In addition, DIM(L) approaches or even surpasses a fully-supervised classifier with similar architecture. DIM with the global-only objective is competitive with some models across tasks, but falls short when compared to generative models and DIM(L) on CIFAR100. Fully-supervised classification results are provided for comparison.

| Model | CIFAR10 | | | CIFAR100 | | |
|---|---|---|---|---|---|---|
| | conv | fc (1024) | $Y(64)$ | conv | fc (1024) | $Y(64)$ |
| Fully supervised | | 75.39 | | | 42.27 | |
| VAE | 60.71 | 60.54 | 54.61 | 37.21 | 34.05 | 24.22 |
| AE | 62.19 | 55.78 | 54.47 | 31.50 | 23.89 | 27.44 |
| $\beta$-VAE | 62.4 | 57.89 | 55.43 | 32.28 | 26.89 | 28.96 |
| AAE | 59.44 | 57.19 | 52.81 | 36.22 | 33.38 | 23.25 |
| BiGAN | 62.57 | 62.74 | 52.54 | 37.59 | 33.34 | 21.49 |
| NAT | 56.19 | 51.29 | 31.16 | 29.18 | 24.57 | 9.72 |
| DIM(G) | 52.2 | 52.84 | 43.17 | 27.68 | 24.35 | 19.98 |
| DIM(L) (DV) | **72.66** | **70.60** | **64.71** | **48.52** | **44.44** | **39.27** |
| DIM(L) (JSD) | **73.25** | **73.62** | **66.96** | **48.13** | **45.92** | **39.60** |
| DIM(L) (infoNCE) | **75.21** | **75.57** | **69.13** | **49.74** | **47.72** | **41.61** |

Table 2: Classification accuracy (top 1) results on Tiny ImageNet and STL-10. For Tiny ImageNet, DIM with the local objective outperforms all other models presented by a large margin, and approaches accuracy of a fully-supervised classifier similar to the Alexnet architecture used here.

| | Tiny ImageNet | | | STL-10 (random crop pretraining) | | | |
|---|---|---|---|---|---|---|---|
| | conv | fc (4096) | $Y(64)$ | conv | fc (4096) | $Y(64)$ | SS |
| Fully supervised | | 36.60 | | | 68.7 | | |
| VAE | 18.63 | 16.88 | 11.93 | 58.27 | 56.72 | 46.47 | 68.65 |
| AE | 19.07 | 16.39 | 11.82 | 58.19 | 55.57 | 46.82 | 70.29 |
| $\beta$-VAE | 19.29 | 16.77 | 12.43 | 57.15 | 55.14 | 46.87 | 70.53 |
| AAE | 18.04 | 17.27 | 11.49 | 59.54 | 54.47 | 43.89 | 64.15 |
| BiGAN | 24.38 | 20.21 | 13.06 | 71.53 | 67.18 | 58.48 | 74.77 |
| NAT | 13.70 | 11.62 | 1.20 | 64.32 | 61.43 | 48.84 | 70.75 |
| DIM(G) | 11.32 | 6.34 | 4.95 | 42.03 | 30.82 | 28.09 | 51.36 |
| DIM(L) (DV) | 30.35 | 29.51 | 28.18 | 69.15 | 63.81 | 61.92 | 71.22 |
| DIM(L) (JSD) | **33.54** | **36.88** | **31.66** | **72.86** | **70.85** | **65.93** | **76.96** |
| DIM(L) (infoNCE) | **34.21** | **38.09** | **33.33** | **72.57** | **70.00** | **67.08** | **76.81** |

CPC and DIM performance improved considerably. We chose a crop size of $25\%$ of the image size in width and depth with a stride of $12.5\%$ the image size (e.g., $8 \times 8$ crops with $4 \times 4$ strides for CIFAR10, $16 \times 16$ crops with $8 \times 8$ strides for STL-10), so that there were a total of $7 \times 7$ local features. For both DIM(L) and CPC, we used infoNCE as well as the same "encode-and-dot-product" architecture (tantamount to a deep bilinear model), rather than the shallow bilinear model used in Oord et al. (2018). For CPC, we used a total of 3 such networks, where each network for CPC is used for a separate prediction task of local feature maps in the next 3 rows of a summary predictor feature within each column.[2] For simplicity, we omitted the prior term, $\beta$, from DIM. Without data augmentation on CIFAR10, CPC performs worse than DIM(L) with a ResNet-50 (He et al., 2016) type architecture. For experiments we ran on STL-10 with data augmentation (using the same encoder architecture as Table 2), CPC and DIM were competitive, with CPC performing slightly better.

CPC makes predictions based on multiple summary features, each of which contains different amounts of information about the full input. We can add similar behavior to DIM by computing *less global* features which condition on $3 \times 3$ blocks of local features sampled at random from the full $7 \times 7$ sets of local features. We then maximize mutual information between these less global features and the full sets of local features. We share a single MI estimator across all possible $3 \times 3$ blocks of local features when using this version of DIM. This represents a particular instance of the occlusion technique described in Section 4.3. The resulting model gave a significant performance boost to

---

[2]Note that this is slightly different from the setup used in Oord et al. (2018), which used a total of 5 such predictors, though we found other configurations performed similarly.

Table 3: Comparisons of DIM with Contrastive Predictive Coding (CPC, Oord et al., 2018). These experiments used a strided-crop architecture similar to the one used in Oord et al. (2018). For CIFAR10 we used a ResNet-50 encoder, and for STL-10 we used the same architecture as for Table 2. We also tested a version of DIM that computes the global representation from a 3x3 block of local features randomly selected from the full 7x7 set of local features. This is a particular instance of the occlusions described in Section 4.3. DIM(L) is competitive with CPC in these settings.

| Model | CIFAR10 (no data augmentation) | STL10 (random crop pretraining) |
|---|---|---|
| DIM(L) single global | 80.95 | 76.97 |
| CPC | 77.45 | 77.81 |
| DIM(L) multiple globals | 77.51 | 78.21 |

Table 4: Extended comparisons on CIFAR10. Linear classification results using SVM are over five runs. MS-SSIM is estimated by training a separate decoder using the fixed representation as input and minimizing the $L2$ loss with the original input. Mutual information estimates were done using MINE and the neural dependence measure (NDM) were trained using a discriminator between unshuffled and shuffled representations.

| Model | Proxies | | | | Neural Estimators | |
|---|---|---|---|---|---|---|
| | SVM (conv) | SVM (fc) | SVM ($Y$) | MS-SSIM | $\widehat{I}_\rho(X, Y)$ | NDM |
| VAE | $53.83 \pm 0.62$ | $42.14 \pm 3.69$ | $39.59 \pm 0.01$ | 0.72 | 93.02 | 1.62 |
| AAE | $55.22 \pm 0.06$ | $43.34 \pm 1.10$ | $37.76 \pm 0.18$ | 0.67 | 87.48 | 0.03 |
| BiGAN | $56.40 \pm 1.12$ | $38.42 \pm 6.86$ | $44.90 \pm 0.13$ | 0.46 | 37.69 | 24.49 |
| NAT | $48.62 \pm 0.02$ | $42.63 \pm 3.69$ | $39.59 \pm 0.01$ | 0.29 | 6.04 | 0.02 |
| DIM(G) | $46.8 \pm 2.29$ | $28.79 \pm 7.29$ | $29.08 \pm 0.24$ | 0.49 | 49.63 | 9.96 |
| DIM(L+G) | $57.55 \pm 1.442$ | $45.56 \pm 4.18$ | $18.63 \pm 4.79$ | 0.53 | 101.65 | 22.89 |
| DIM(L) | $63.25 \pm 0.86$ | $54.06 \pm 3.6$ | $49.62 \pm 0.3$ | 0.37 | 45.09 | 9.18 |

DIM for STL-10. Surprisingly, this same architecture performed worse than using the fully global representation with CIFAR10. Overall DIM only slightly outperforms CPC in this setting, which suggests that the strictly ordered autoregression of CPC may be unnecessary for some tasks.

**Extended comparisons** Tables 4 shows results on linear separability, reconstruction (MS-SSIM), mutual information, and dependence (NDM) with the CIFAR10 dataset. We did not compare to CPC due to the divergence of architectures. For linear classifier results (SVC), we trained five support vector machines with a simple hinge loss for each model, averaging the test accuracy. For MINE, we used a decaying learning rate schedule, which helped reduce variance in estimates and provided faster convergence.

MS-SSIM correlated well with the MI estimate provided by MINE, indicating that these models encoded pixel-wise information well. Overall, all models showed much lower dependence than BiGAN, indicating the marginal of the encoder output is not matching to the generator's spherical Gaussian input prior, though the mixed local/global version of DIM is close. For MI, reconstruction-based models like VAE and AAE have high scores, and we found that combining local and global DIM objectives had very high scores ($\alpha = 0.5$, $\beta = 0.1$ is presented here as DIM(L+G)). For more in-depth analyses, please see the ablation studies and the nearest-neighbor analysis in the App. (A.4, A.5).

## 4.3 ADDING COORDINATE INFORMATION AND OCCLUSIONS

Maximizing MI between global and local features is not the only way to leverage image structure. We consider augmenting DIM by adding input occlusion when computing global features and by adding auxiliary tasks which maximize MI between local features and absolute or relative spatial coordinates given a global feature. These additions improve classification results (see Table 5).

For occlusion, we randomly occlude part of the input when computing the global features, but compute local features using the full input. Maximizing MI between occluded global features and unoccluded local features aggressively encourages the global features to encode information which is shared across the entire image. For coordinate prediction, we maximize the model's ability to predict the coordinates $(i, j)$ of a local feature $c_{(i,j)} = C_\psi^{(i,j)}(x)$ after computing the global features

Table 5: Augmenting infoNCE DIM with additional structural information – adding coordinate prediction tasks or occluding input patches when computing the global feature vector in DIM can improve the classification accuracy, particularly with the highly-compressed global features.

| Model | CIFAR10 | | | CIFAR100 | | |
|---|---|---|---|---|---|---|
| | $Y(64)$ | fc (1024) | conv | $Y(64)$ | fc (1024) | conv |
| DIM | 70.65 | 73.33 | 77.46 | 44.27 | 47.96 | 49.90 |
| DIM (coord) | 71.56 | 73.89 | 77.28 | 45.37 | 48.61 | 50.27 |
| DIM (occlude) | 72.87 | 74.45 | 76.77 | 44.89 | 47.65 | 48.87 |
| DIM (coord + occlude) | 73.99 | 75.15 | 77.27 | 45.96 | 48.00 | 48.72 |

$y = E_\psi(x)$. To accomplish this, we maximize $\mathbb{E}[\log p_\theta((i,j)|y, c_{(i,j)})]$ (i.e., minimize the cross-entropy). We can extend the task to maximize conditional MI given global features $y$ between pairs of local features $(c_{(i,j)}, c_{(i',j')})$ and their relative coordinates $(i - i', j - j')$. This objective can be written as $\mathbb{E}[\log p_\theta((i - i', j - j')|y, c_{(i,j)}, c_{(i',j')})]$. We use both these objectives in our results.

Additional implementation details can be found in the App. (A.7). Roughly speaking, our input occlusions and coordinate prediction tasks can be interpreted as generalizations of inpainting (Pathak et al., 2016) and context prediction (Doersch et al., 2015) tasks which have previously been proposed for self-supervised feature learning. Augmenting DIM with these tasks helps move our method further towards learning representations which encode images (or other types of inputs) not just in terms of compressing their low-level (e.g. pixel) content, but in terms of distributions over relations among higher-level features extracted from their lower-level content.

## 5 CONCLUSION

In this work, we introduced Deep InfoMax (DIM), a new method for learning unsupervised representations by maximizing mutual information, allowing for representations that contain locally-consistent information across structural "locations" (e.g., patches in an image). This provides a straightforward and flexible way to learn representations that perform well on a variety of tasks. We believe that this is an important direction in learning higher-level representations.

## 6 ACKNOWLEDGEMENTS

RDH received partial support from IVADO, NIH grants 2R01EB005846, P20GM103472, P30GM122734, and R01EB020407, and NSF grant 1539067. AF received partial support from NIH grants R01EB020407, R01EB006841, P20GM103472, P30GM122734. We would also like to thank Geoff Gordon (MSR), Ishmael Belghazi (MILA), Marc Bellemare (Google Brain), Mikołaj Bińkowski (Imperial College London), Simon Sebbagh, and Aaron Courville (MILA) for their useful input at various points through the course of this research.

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

# A APPENDIX

## A.1 ON THE JENSEN-SHANNON DIVERGENCE AND MUTUAL INFORMATION

Here we show the relationship between the Jensen-Shannon divergence (JSD) between the joint and the product of marginals and the pointwise mutual information (PMI). Let $p(x)$ and $p(y)$ be two marginal densities, and define $p(y|x)$ and $p(x,y) = p(y|x)p(x)$ as the conditional and joint distribution, respectively. Construct a probability mixture density, $m(x,y) = \frac{1}{2}(p(x)p(y) + p(x,y))$. It follows that $m(x) = p(x)$, $m(y) = p(y)$, and $m(y|x) = \frac{1}{2}(p(y) + p(y|x))$.

Note that:

$$\log m(y|x) = \log\left(\tfrac{1}{2}(p(y) + p(y|x))\right) = \log\frac{1}{2} + \log p(y) + \log\left(1 + \tfrac{p(y|x)}{p(y)}\right). \qquad (9)$$

Discarding some constants:

$JSD(p(x,y)||p(x)p(y))$

$$\propto \mathbb{E}_{x\sim p(x)}\left[\mathbb{E}_{y\sim p(y|x)}\left[\log\frac{p(y|x)p(x)}{m(y|x)m(x)}\right] + \mathbb{E}_{y\sim p(y)}\left[\log\frac{p(y)p(x)}{m(y|x)m(x)}\right]\right]$$

$$\propto \mathbb{E}_{x\sim p(x)}\left[\mathbb{E}_{y\sim p(y|x)}\left[\log\frac{p(y|x)}{p(y)} - \log\left(1 + \tfrac{p(y|x)}{p(y)}\right)\right] + \mathbb{E}_{y\sim p(y)}\left[-\log\left(1 + \tfrac{p(y|x)}{p(y)}\right)\right]\right]$$

$$\propto \mathbb{E}_{x\sim p(x)}\left[\mathbb{E}_{y\sim p(y|x)}\left[\log\frac{p(y|x)}{p(y)}\right] - 2\mathbb{E}_{y\sim m(y|x)}\left[\log\left(1 + \tfrac{p(y|x)}{p(y)}\right)\right]\right]$$

$$\propto \mathbb{E}_{x\sim p(x)}\left[\mathbb{E}_{y\sim p(y|x)}\left[\log\frac{p(y|x)}{p(y)} - 2\tfrac{m(y|x)}{p(y|x)}\log\left(1 + \tfrac{p(y|x)}{p(y)}\right)\right]\right]$$

$$\propto \mathbb{E}_{x\sim p(x)}\left[\mathbb{E}_{y\sim p(y|x)}\left[\log\frac{p(y|x)}{p(y)} - (1 + \tfrac{p(y)}{p(y|x)})\log\left(1 + \tfrac{p(y|x)}{p(y)}\right)\right]\right]. \qquad (10)$$

The quantity inside the expectation of Eqn. 10 is a concave, monotonically increasing function of the ratio $\frac{p(y|x)}{p(y)}$, which is exactly $e^{\text{PMI}(x,y)}$. Note this relationship does not hold for the JSD of arbitrary distributions, as the the joint and product of marginals are intimately coupled.

We can verify our theoretical observation by plotting the JSD and KL divergences between the joint and the product of marginals, the latter of which is the formal definition of mutual information (MI). As computing the continuous MI is difficult, we assume a discrete input with uniform probability, $p(x)$ (e.g., these could be one-hot variables indicating one of $N$ i.i.d. random samples), and a randomly initialized $N \times M$ joint distribution, $p(x,y)$, such that $\sum_{j=1}^{M} p(x_i, y_j) = 1\ \forall i$. For this joint distribution, we sample from a uniform distribution, then apply dropout to encourage sparsity to simulate the situation when there is no bijective function between $x$ and $y$, then apply a softmax. As the distributions are discrete, we can compute the KL and JSD between $p(x,y)$ and $p(x)p(y)$.

We ran these experiments with matched input / output dimensions of 8, 16, 32, 64, and 128, randomly drawing 1000 joint distributions, and computed the KL and JSD divergences directly. Our results (Figure A.1) indicate that the KL (traditional definition of mutual information) and the JSD have an approximately monotonic relationship. Overall, the distributions with the highest mutual information also have the highest JSD.

## A.2 EXPERIMENT AND ARCHITECTURE DETAILS

Here we provide architectural details for our experiments. Example code for running Deep Infomax (DIM) can be found at https://github.com/rdevon/DIM.

**Encoder** We used an encoder similar to a deep convolutional GAN (DCGAN, Radford et al., 2015) discriminator for CIFAR10 and CIFAR100, and for all other datasets we used an Alexnet (Krizhevsky et al., 2012) architecture similar to that found in Donahue et al. (2016). ReLU activations and batch norm (Ioffe & Szegedy, 2015) were used on every hidden layer. For the DCGAN architecture, a single hidden layer with 1024 units was used after the final convolutional layer, and for the Alexnet architecture it was two hidden layers with 4096. For all experiments, the output of all encoders was a 64 dimensional vector.

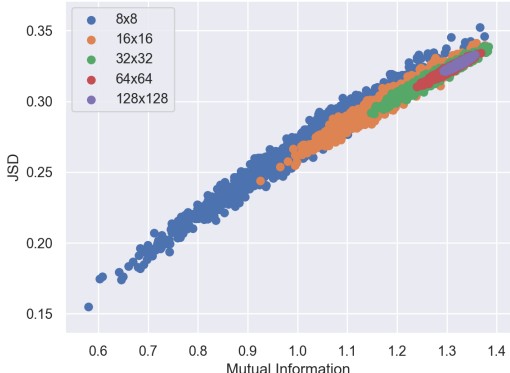

Figure 4: Scatter plots of $MI(x;y)$ versus $JSD(p(x,y)\|p(x)p(y))$ with discrete inputs and a given randomized and sparse joint distribution, $p(x,y)$. $8 \times 8$ indicates a square joint distribution with 8 rows and 8 columns. Our experiments indicate a strong monotonic relationship between $MI(x;y)$ and $JSD(p(x,y)\|p(x)p(y))$ Overall, the distributions with the highest $MI(x;y)$ have the highest $JSD(p(x,y)\|p(x)p(y))$.

**Mutual information discriminators**  For the global mutual information objective, we first encode the input into a feature map, $C_\psi(x)$, which in this case is the output of the last convolutional layer. We then encode this representation further using linear layers as detailed above to get $E_\psi(x)$. $C_\psi(x)$ is then flattened, then concatenated with $E_\psi(x)$. We then pass this to a fully-connected network with two 512-unit hidden layers (see Table 6).

Table 6: Global DIM network architecture

| Operation | Size | Activation |
|---|---|---|
| Input $\rightarrow$ Linear layer | 512 | ReLU |
| Linear layer | 512 | ReLU |
| Linear layer | 1 | |

We tested two different architectures for the local objective. The first (Figure 5) concatenated the global feature vector with the feature map at every location, i.e., $\{[C_\psi^{(i)}(x), E_\psi(x)]\}_{i=1}^{M \times M}$. A $1 \times 1$ convolutional discriminator is then used to score the (feature map, feature vector) pair,

$$T_{\psi,\omega}^{(i)}(x, y(x)) = D_\omega([C_\psi^{(i)}(x), E_\psi(x)]). \tag{11}$$

Fake samples are generated by combining global feature vectors with local feature maps coming from different images, $x'$:

$$T_{\psi,\omega}^{(i)}(x', E_\psi(x)) = D_\omega([C_\psi^{(i)}(x'), E_\psi(x)]). \tag{12}$$

This architecture is featured in the results of Table 4, as well as the ablation and nearest-neighbor studies below. We used a $1 \times 1$ convnet with two 512-unit hidden layers as discriminator (Table 7).

Table 7: Local DIM concat-and-convolve network architecture

| Operation | Size | Activation |
|---|---|---|
| Input $\rightarrow 1 \times 1$ conv | 512 | ReLU |
| $1 \times 1$ conv | 512 | ReLU |
| $1 \times 1$ conv | 1 | |

The other architecture we tested (Figure 6) is based on non-linearly embedding the global and local features in a (much) higher-dimensional space, and then computing pair-wise scores using dot products between their high-dimensional embeddings. This enables efficient evaluation of a large number of pair-wise scores, thus allowing us to use large numbers of positive/negative samples. Given a sufficiently high-dimensional embedding space, this approach can represent (almost) arbitrary classes of pair-wise functions that are non-linear in the original, lower-dimensional features. For more information, refer to Reproducing Kernel Hilbert Spaces. We pass the global feature through a

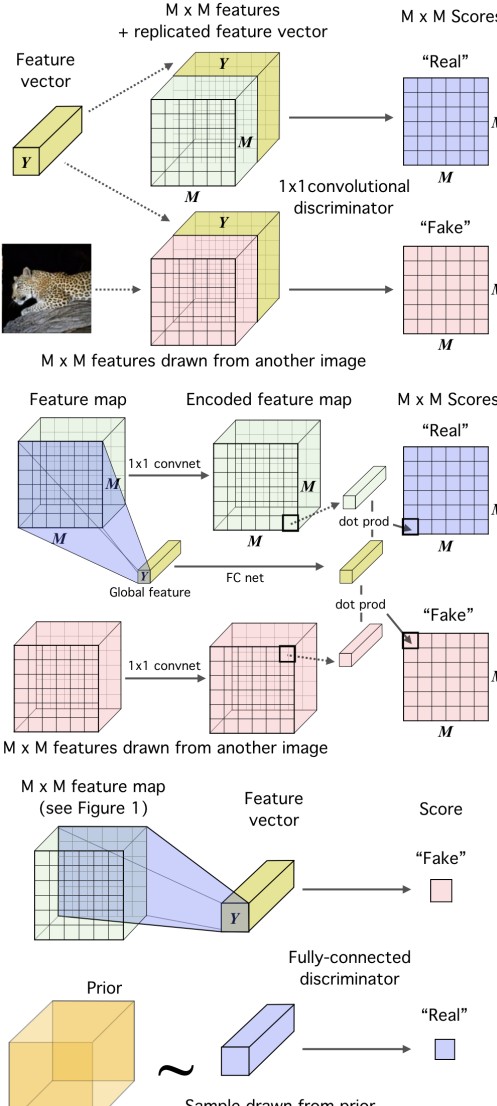

Figure 5: **Concat-and-convolve architecture.** The global feature vector is concatenated with the lower-level feature map *at every location*. A $1 \times 1$ convolutional discriminator is then used to score the "real" feature map / feature vector pair, while the "fake" pair is produced by pairing the feature vector with a feature map from another image.

Figure 6: **Encode-and-dot-product architecture.** The global feature vector is encoded using a fully-connected network, and the lower-level feature map is encoded using 1x1 convolutions, but with the same number of output features. We then take the dot-product between the feature at each location of the feature map encoding and the encoded global vector for scores.

Figure 7: **Matching the output of the encoder to a prior.** "Real" samples are drawn from a prior while "fake" samples from the encoder output are sent to a discriminator. The discriminator is trained to distinguish between (classify) these sets of samples. The encoder is trained to "fool" the discriminator.

fully connected neural network to get the encoded global feature, $S_\omega(E_\psi(x))$. In our experiments, we used a single hidden layer network with a linear shortcut (See Table 8).

Table 8: Local DIM encoder-and-dot architecture for global feature

| Operation | Size | Activation | Output |
|---|---|---|---|
| Input $\rightarrow$ Linear | 2048 | ReLU | |
| Linear | 2048 | | Output 1 |
| Input $\rightarrow$ Linear | 2048 | ReLU | Output 2 |
| Output 1 + Output 2 | | | |

We embed each local feature in the local feature map $C_\psi(x)$ using an architecture which matches the one for global feature embedding. We apply it via $1 \times 1$ convolutions. Details are in Table 9.

Table 9: Local DIM encoder-and-dot architecture for local features

| Operation | Size | Activation | Output |
|---|---|---|---|
| Input $\rightarrow 1 \times 1 conv$ | 2048 | ReLU | |
| $1 \times 1$ conv | 2048 | | Output 1 |
| Input $\rightarrow 1 \times 1$ conv | 2048 | ReLU | Output 2 |
| Output 1 + Output 2 | | | |
| Block Layer Norm | | | |

Finally, the outputs of these two networks are combined by matrix multiplication, summing over the feature dimension ($2048$ in the example above). As this is computed over a batch, this allows us to efficiently compute both positive and negative examples simultaneously. This architecture is featured in our main classification results in Tables 1, 2, and 5.

For the local objective, the feature map, $C_\psi(x)$, can be taken from any level of the encoder, $E_\psi$. For the global objective, this is the last convolutional layer, and this objective was insensitive to which layer we used. For the local objectives, we found that using the next-to-last layer worked best for CIFAR10 and CIFAR100, while for the other larger datasets it was the previous layer. This sensitivity is likely due to the relative size of the of the receptive fields, and further analysis is necessary to better understand this effect. Note that all feature maps used for DIM included the final batch normalization and ReLU activation.

**Prior matching**  Figure 7 shows a high-level overview of the prior matching architecture. The discriminator used to match the prior in DIM was a fully-connected network with two hidden layers of 1000 and 200 units (Table 10).

Table 10: Prior matching network architecture

| Operation | Size | Activation |
|---|---|---|
| Input $\rightarrow$ Linear layer | 1000 | ReLU |
| Linear layer | 200 | ReLU |
| Linear layer | 1 | |

**Generative models**  For generative models, we used a similar setup as that found in Donahue et al. (2016) for the generators / decoders, where we used a generator from DCGAN in all experiments.

All models were trained using Adam with a learning rate of $1 \times 10^{-4}$ for 1000 epochs for CIFAR10 and CIFAR100 and for 200 epochs for all other datasets.

**Contrastive Predictive Coding**  For Contrastive Predictive Coding (CPC, Oord et al., 2018), we used a simple a GRU-based PixelRNN (Oord et al., 2016) with the same number of hidden units as the feature map depth. All experiments with CPC had the global state dimension matched with the size of these recurrent hidden units.

## A.3  SAMPLING STRATEGIES

We found both infoNCE and the DV-based estimators were sensitive to negative sampling strategies, while the JSD-based estimator was insensitive. JSD worked better ($1-2\%$ accuracy improvement) by excluding positive samples from the product of marginals, so we exclude them in our implementation. It is quite likely that this is because our batchwise sampling strategy overestimate the frequency of positive examples as measured across the complete dataset. infoNCE was highly sensitive to the number of negative samples for estimating the log-expectation term (see Figure 9). With high sample size, infoNCE outperformed JSD on many tasks, but performance drops quickly as we reduce the number of images used for this estimation. This may become more problematic for larger datasets and networks where available memory is an issue. DV was outperformed by JSD even with the maximum number of negative samples used in these experiments, and even worse was highly unstable as the number of negative samples dropped.

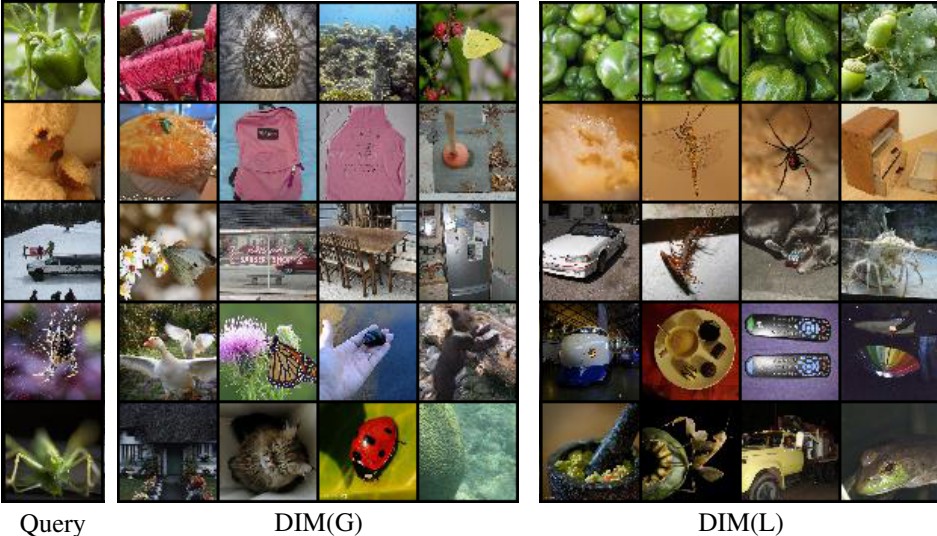

Query              DIM(G)              DIM(L)

Figure 8: Nearest-neighbor using the $L_1$ distance on the encoded Tiny ImageNet images, with DIM(G) and DIM(L). The images on the far left are randomly-selected reference images (query) from the training set and the four images their nearest-neighbor from the test set as measured in the representation, sorted by proximity. The nearest neighbors from DIM(L) are much more interpretable than those with the purely global objective.

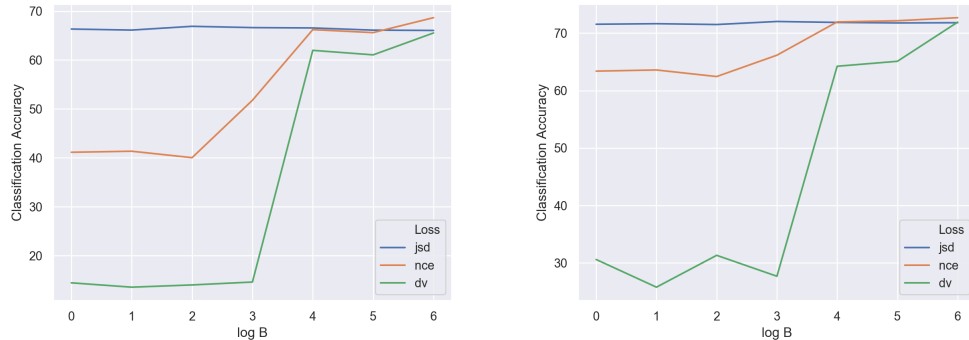

Figure 9: Classification accuracies (left: global representation, $Y$, right: convolutional layer) for CIFAR10, first training DIM, then training a classifier for 1000 epochs, keeping the encoder fixed. Accuracies shown averaged over the last 100 epochs, averaged over 3 runs, for the infoNCE, JSD, and DV DIM losses. x-axis is log base-2 of the number of negative samples (0 mean one negative sample per positive sample). JSD is insensitive to the number of negative samples, while infoNCE shows a decline as the number of negative samples decreases. DV also declines, but becomes unstable as the number of negative samples becomes too low.

## A.4 NEAREST-NEIGHBOR ANALYSIS

In order to better understand the metric structure of DIM's representations, we did a nearest-neighbor analysis, randomly choosing a sample from each class in the test set, ordering the test set in terms of $L_1$ distance in the representation space (to reflect the uniform prior), then selecting the four with the lowest distance. Our results in Figure 8 show that DIM with a local-only objective, DIM(L), learns a representation with a much more interpretable structure across the image. However, our result potentially highlights an issue with using only consistent information across patches, as many of the nearest neighbors share patterns (colors, shapes, texture) but not class.

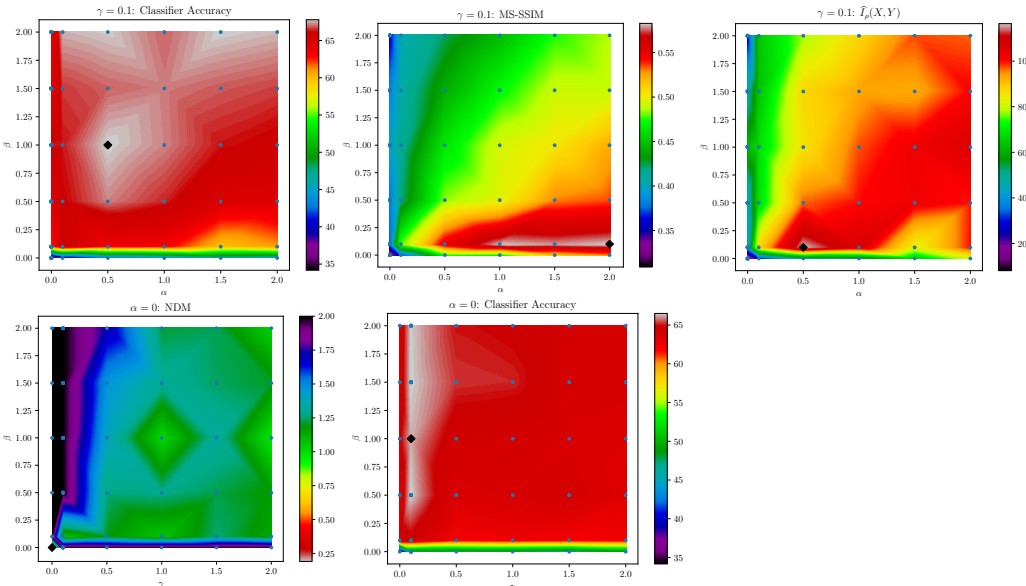

Figure 10: Results from the ablation studies with DIM on CIFAR10. Values calculated are points on the grid, and the heatmaps were derived by bilinear interpolation. Heatmaps were thresholded at the minimum value (or maximum for NDM) for visual clarity. Highest (or lowest) value is marked on the grid. NDM here was measured without the sigmoid function.

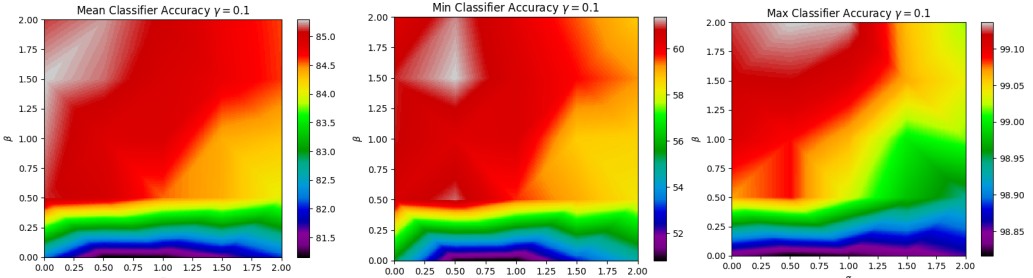

Figure 11: Ablation study on CelebA over the global and local parameters, $\alpha$ and $\beta$. The classification task is multinomial, so provided is the average, minimum, and maximum class accuracies across attibutes. While the local objective is crucial, the global objective plays a stronger role here than with other datasets.

## A.5  ABLATION STUDIES

To better understand the effects of hyperparameters $\alpha$, $\beta$, and $\gamma$ on the representational characteristics of the encoder, we performed several ablation studies. These illuminate the relative importance of global verses local mutual information objectives as well as the role of the prior.

**Local versus global mutual information maximization**   The results of our ablation study for DIM on CIFAR10 are presented in Figure 10. In general, good classification performance is highly dependent on the local term, $\beta$, while good reconstruction is highly dependent on the global term, $\alpha$. However, a small amount of $\alpha$ helps in classification accuracy and a small about of $\beta$ improves reconstruction. For mutual information, we found that having a combination of $\alpha$ and $\beta$ yielded higher MINE estimates. Finally, for CelebA (Figure 11), where the classification task is more fine-grained (is composed of potentially locally-specified labels, such as "lipstick" or "smiling"), the global objective plays a stronger role than with classification on other datasets (e.g., CIFAR10).

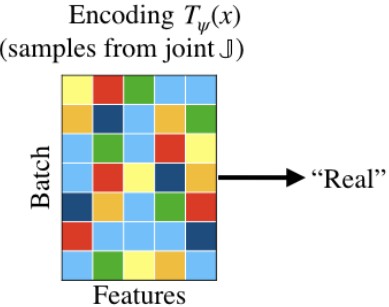

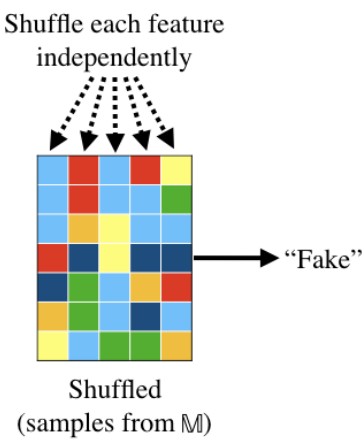

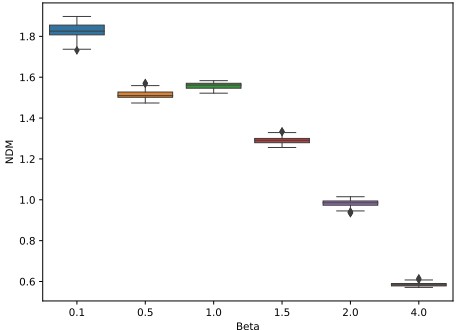

Figure 12: A schematic of learning the Neural Dependency Measure. For a given batch of inputs, we encode this into a set of representations. We then shuffle each feature (dimension of the feature vector) across the batch axis. The original version is sent to the discriminator and given the label "real", while the shuffled version is labeled as "fake". The easier this task, the more dependent the components of the representation.

Figure 13: Neural Dependency Measures (NDMs) for various $\beta$-VAE (Alemi et al., 2016; Higgins et al., 2016) models $(0.1, 0.5, 1.0, 1.5, 2.0, 4.0)$. Error bars are provided over five runs of each VAE and estimating NDM with 10 different networks. We find that there is a strong trend as we increase the value of $\beta$ and that the estimates are relatively consistent and informative w.r.t. independence as expected.

**The effect of the prior** We found including the prior term, $\gamma$, was absolutely necessary for ensuring low dependence between components of the high-level representation, $E_\psi(x)$, as measured by NDM. In addition, a small amount of the prior term helps improve classification results when used with the local term, $\beta$. This may be because the additional constraints imposed on the representation help to encourage the local term to focus on consistent, rather than trivial, information.

## A.6 EMPIRICAL CONSISTENCY OF NEURAL DEPENDENCY MEASURE (NDM)

Here we evaluate the Neural Dependency Measure (NDM) over a range of $\beta$-VAE (Alemi et al., 2016; Higgins et al., 2016) models. $\beta$-VAE encourages disentangled representations by increasing the role of the KL-divergence term in the ELBO objective. We hypthesized that NDM would consistently measure lower dependence (lower NDM) as the $\beta$ values increase, and our results in Figure A.6 confirm this. As we increase $\beta$, there is a strong downward trend in the NDM, though $\beta = 0.5$ and

$\beta = 1.0$ give similar numbers. In addition, the variance over estimates and models is relatively low, meaning the estimator is empirically consistent in this setting.

## A.7 ADDITIONAL DETAILS ON OCCLUSION AND COORDINATE PREDICTION EXPERIMENTS

Here we present experimental details on the occlusion and coordinate prediction tasks.

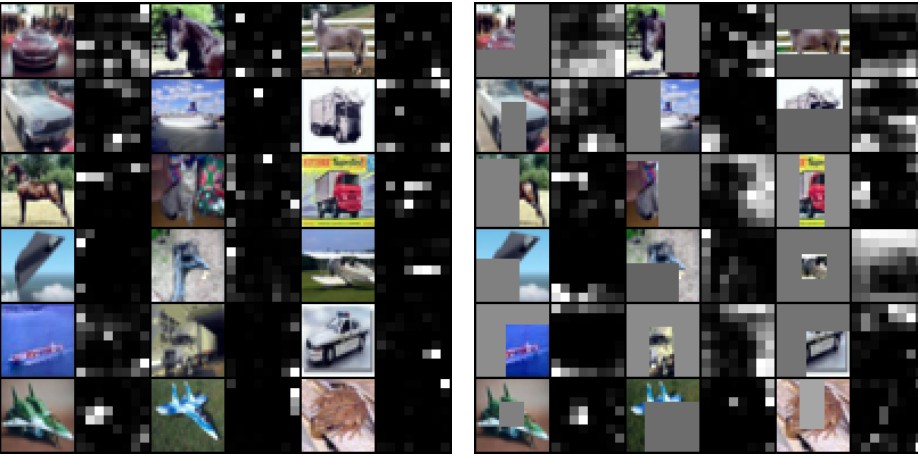

<p align="center">Training without occlusion       Training with occlusion</p>

Figure 14: Visualizing model behaviour when computing global features with and without occlusion, for NCE-based DIM. The images in each block of images come in pairs. The left image in each pair shows the model input when computing the global feature vector. The right image shows the NCE loss suffered by the score between that global feature vector and the local feature vector at each location in the $8 \times 8$ local feature map computed from the unoccluded image. This loss is equal to minus the value in Equation 5. With occluded inputs, this loss tends to be highest for local features with receptive fields that overlap the occluded region.

**Occlusions.** For the occlusion experiments, the sampling distribution for patches to occlude was ad-hoc. Roughly, we randomly occlude the input image under the constraint that at least one $10 \times 10$ block of pixels remains visible and at least one $10 \times 10$ block of pixels is fully occluded. We chose $10 \times 10$ based on the receptive fields of local features in our encoder, since it guarantees that occlusion leaves at least one local feature fully observed and at least one local feature fully unobserved. Figure 14 shows the distribution of occlusions used in our tests.

**Absolute coordinate prediction** For absolute coordinate prediction, the global features $y$ and local features $c_{(i,j)}$ are sampled by 1) feeding an image from the data distribution through the feature encoder, and 2) sampling a random spatial location $(i, j)$ from which to take the local features $c_{(i,j)}$. Given $y$ and $c_{(i,j)}$, we treat the coordinates $i$ and $j$ as independent categorical variables and measure the required log probability using a sum of categorical cross-entropies. In practice, we implement the prediction function $p_\theta$ as an MLP with two hidden layers, each with 512 units, ReLU activations, and batchnorm. We marginalize this objective over all local features associated with a given global feature when computing gradients.

**Relative coordinate prediction** For relative coordinate prediction, the global features $y$ and local features $c_{(i,j)}/c_{(i',j')}$ are sampled by 1) feeding an image from the data distribution through the feature encoder, 2) sampling a random spatial location $(i, j)$ from which to take *source* local features $c_{(i,j)}$, and 3) sampling another random location $(i', j')$ from which to take *target* local features $c_{(i',j')}$. In practice, our predictive model for this task uses the same architecture as for the task described previously. For each global feature $y$ we select one source feature $c_{(i,j)}$ and marginalize over all possible target features $c_{(i',j')}$ when computing gradients.

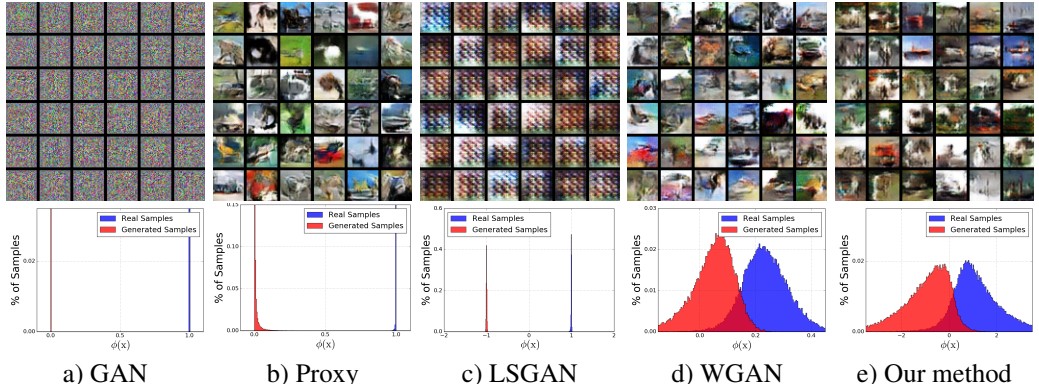

a) GAN      b) Proxy      c) LSGAN      d) WGAN      e) Our method

Figure 15: Histograms depicting the discriminator's unnormalized output distribution for the standard GAN, GAN with $-\log D$ loss, Least Squares GAN, Wasserstein GAN and our proposed method when trained with a 50:1 training ratio.

## A.8 TRAINING A GENERATOR BY MATCHING TO A PRIOR IMPLICITLY

We show here and in our experiments below that we can use prior objective in DIM (Equation 7) to train a high-quality generator of images by training $\mathbb{U}_{\psi,\mathbb{P}}$ to map to a one-dimensional mixture of two Gaussians implicitly. One component of this mixture will be a target for the push-forward distribution of $\mathbb{P}$ through the encoder while the other will be a target for the push-forward distribution of the generator, $\mathbb{Q}_\theta$, through the same encoder.

Let $G_\theta : \mathcal{Z} \to \mathcal{X}$ be a generator function, where the input $z \in \mathcal{Z}$ is drawn from a simple prior, $p(z)$ (such as a spherical Gaussian). Let $\mathbb{Q}_\theta$ be the generated distribution and $\mathbb{P}$ be the empirical distribution of the training set. Like in GANs, we will pass the samples of the generator or the training data through another function, $E_\psi$, in order to get gradients to find the parameters, $\theta$. However, unlike GANs, we will not play the minimax game between the generator and this function. Rather $E_\psi$ will be trained to *generate* a mixture of Gaussians conditioned on whether the input sample came from $\mathbb{P}$ or $\mathbb{Q}_\theta$:

$$\mathbb{V}_\mathbb{P} = \mathcal{N}(\mu_P, 1), \quad \mathbb{V}_\mathbb{Q} = \mathcal{N}(\mu_Q, 1), \quad \mathbb{U}_{\psi,\mathbb{P}} = \mathbb{P}\#E_\psi, \quad \mathbb{U}_{\psi,\mathbb{Q}} = \mathbb{Q}_\theta\#E_\psi, \quad (13)$$

where $\mathcal{N}(\mu_P, 1)$ and $\mathcal{N}(\mu_Q, 1)$ are normal distributions with unit variances and means $\mu_P$ and $\mu_Q$ respectively. In order to find the parameters $\psi$, we introduce two discriminators, $T^P_{\phi_P}, T^Q_{\phi_Q} : \mathcal{Y} \to \mathbb{R}$, and use the lower bounds following defined by the JSD f-GAN:

$$(\hat{\psi}, \hat{\phi}_P, \hat{\phi}_Q) = \arg\min_\psi \arg\max_{\phi_P, \phi_Q} \mathcal{L}_d(\mathbb{V}_\mathbb{P}, \mathbb{U}_{\psi,\mathbb{P}}, T^P_{\phi_P}) + \mathcal{L}_d(\mathbb{V}_\mathbb{Q}, \mathbb{U}_{\psi,\mathbb{Q}}, T^Q_{\phi_Q}). \quad (14)$$

The generator is trained to move the first-order moment of $\mathbb{E}_{\mathbb{U}_{\psi,\mathbb{Q}}}[y] = \mathbb{E}_{p(z)}[E_\psi(G_\theta(z))]$ to $\mu_P$:

$$\hat{\theta} = \arg\min(\mathbb{E}_{p(z)}[E_\psi(G_\theta(z))] - \mu_P)^2. \quad (15)$$

Some intuition might help understand why this might work. As discussed in Arjovsky & Bottou (2017), if $\mathbb{P}$ and $\mathbb{Q}_\theta$ have support on a low-dimensional manifolds on $\mathcal{X}$, unless they are perfectly aligned, there exists a discriminator that will be able to perfectly distinguish between samples coming from $\mathbb{P}$ and $\mathbb{Q}_\theta$, which means that $\mathbb{U}_{\psi,\mathbb{P}}$ and $\mathbb{U}_{\psi,\mathbb{Q}}$ must also be disjoint.

However, to train the generator, $\mathbb{U}_{\psi,\mathbb{P}}$ and $\mathbb{U}_{\psi,\mathbb{Q}}$ need to share support on $\mathcal{Y}$ in order to ensure stable and non-zero gradients for the generator. Our own experiments by overtraining the discriminator (Figure 15) confirm that lack of overlap between the two modes of the discriminator is symptomatic of poor training.

Suppose we start with the assumption that the encoder targets, $\mathbb{V}_\mathbb{P}$ and $\mathbb{V}_\mathbb{Q}$, should overlap. Unless $\mathbb{P}$ and $\mathbb{Q}_\theta$ are perfectly aligned (which according to Arjovsky & Bottou (2017) is almost guaranteed not to happen with natural images), then the discriminator can always accomplish this task by discarding information about $\mathbb{P}$ or $\mathbb{Q}_\theta$. This means that, by choosing the overlap, we fix the strength of the encoder.

Table 11: Generation scores on the Tiny Imagenet dataset for non-saturating GAN with contractive penalty (NS-GAN-CP), Wasserstein GAN with gradient penalty (WGAN-GP) and our method. Our encoder was penalized using CP.

| Model | Inception score | FID |
|---|---|---|
| Real data | $31.21 \pm .68$ | 4.03 |
| IE (ours) | $7.41 \pm .10$ | 55.15 |
| NS-GAN-CP | $8.65 \pm .08$ | 40.17 |
| WGAN-GP | $8.38 \pm 0.18$ | 42.30 |

### A.9 GENERATION EXPERIMENTS AND RESULTS

For the generator and encoder, we use a ResNet architecture (He et al., 2016) identical to the one found in Gulrajani et al. (2017). We used the contractive penalty (found in Mescheder et al. (2018) but first introduced in contractive autoencoders (Rifai et al., 2011)) on the encoder, gradient clipping on the discriminators, and no regularization on the generator. Batch norm (Ioffe & Szegedy, 2015) was used on the generator, but not on the discriminator. We trained on $64 \times 64$ dimensional LSUN (Yu et al., 2015), CelebA (Liu et al., 2015), and Tiny Imagenet dataset.

### A.10 IMAGES GENERATION

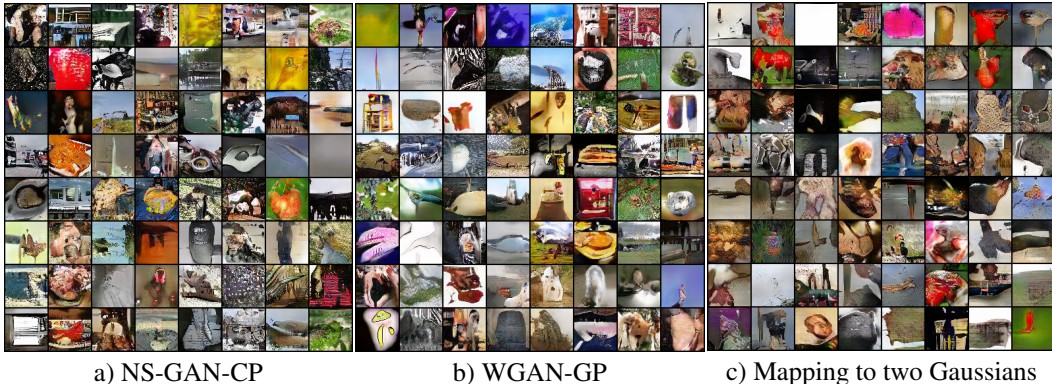

a) NS-GAN-CP      b) WGAN-GP      c) Mapping to two Gaussians

Figure 16: Samples of generated results used to get scores in Table 11. For every methods, the sample are generated after 100 epochs and the models are the same. Qualitative results from these three methods show no qualitative difference.

Here, we train a generator mapping to two Gaussian implicitly as described in Section A.8. Our results (Figure 16) show highly realistic images qualitatively competitive to other methods (Gulrajani et al., 2017; Hjelm et al., 2018). In order to quantitatively compare our method to GANs, we trained a non-saturating GAN with contractive penalty (NS-GAN-CP) and WGAN-GP (Gulrajani et al., 2017) with identical architectures and training procedures. Our results (Table 11) show that, while our mehtod did not surpass NS-GAN-CP or WGAN-GP in our experiments, they came reasonably close.

