# OpenReview forum: "Learning deep representations by mutual information estimation and maximization"
_ICLR.cc/2019/Conference_

### Official Review · AnonReviewer2 · 2018-10-24
**Possibly important paper.**

**Rating:** 9
**Confidence:** 3

**Review:**

Revision 2: The new comparisons with CPC are very helpful.  Most of my other comments are addressed in the response and paper revision.  I am still uncomfortable with the sentence "Our method ... compares favorably with fully-supervised learning on several classification tasks in the settings studied."  This strongly suggests to me that you are claiming to be competitive with SOTA supervised methods.  The paper does not contain supervised results for the resnet-50 architecture.  I would recommend that this sentence should either be dropped from the abstract or have the phrase "in the settings studied" replaced by "for an alexnet architecture".  If you have supervised results for resnet-50 they should be added to table 3 and the abstract could be adjusted to that.  I apologize that this is coming after the update deadline (I have been traveling).  The authors should simply consider the reaction of the community to over-claiming.  Because of the new comparisons with CPC on resnet-50 I am upping my score.  My confidence is low only because the real significance can only be judged over time.

Revision 1: This is a revision of my earlier review.  My overly-excited earlier rating was based on tables 1 and 2 and the claim to have unsupervised features that are competitive with fully-supervised features. (I also am subject to an a-priori bias in favor of mutual information methods.)  I took the authors word for their claim and submitted the review without investigating existing results on CIFAR10.  It seems that tables 1 and 2 are presenting extremely weak fully supervised baselines.  If DIM(L) can indeed produce features that are competitive with state of the art fully supervised features, the result is extremely important.  But this claim seems misrepresented in the paper.

Original review:

There is a lot of material in this paper and I respect this groups
high research-to-publication ratio. However, it might be nice to have
the paper more focused on the subset of ideas that seem to matter.

My biggest comment is that the top level spin seems wrong.
Specifically, the paper focuses on the two bullets on page 3 ---
mutual information and statistical constraints.  Here mutual
information is interpreted as the information between the input and
output of a feature encoder.  Clearly this has a trivial solution
where the input equals the output so the second bullet --- statistical
constraints --- are required.  But the empirical content of the paper
strongly undermines these top level bullets.  Setting the training
objective to be the a balance of MI between input and output under a
statistical consrtraint leads to DIM(G) which, according the results in
the paper, is an empirical disaster.  DIM(L) is the main result and
something else seems to be going on there (more later).  Furthermore,
the empirical results suggest that the second bullet --- statistical
constraints --- is of very little value for DIM(L). The key ablation
study here seems to be missing from the paper.  Appendix A.4 states
that "a small amount of the [statistical constraint] helps improve
classification results when used with the [local information
objective].  No quantitative ablation number is given.  Other measures
of the statistical constraint seem to simply measure to what extent
the constraint has been successfully enforced.  But the results
suggest that even successfully enforcing the constraint is of little,
if any, value for the ability of the features to be effective in
prediction.  So, it seems to me, the paper to really just about the
local information objective.

The real power house of the paper --- the local information objective
--- seems related to mutual information predictive coding as
formalized in the recent paper from deep mind by van den Oord et al
and also an earlier arxiv paper by McAllester on information-theoretic
co-training.  In these other papers one assumes a signal x_1, ... x_T
and tries to extract low dimensional features F(x_t) such that F(x_1),
..., F(x_t) carries large mutual information with F(x_{t+1}).  The
local objective of this paper takes a signal x1, ..., x_k (nXn
subimages) and extracts local features F(x_1), ... F(x_k) and a global
feature Y(F(x_1), ..., F(x_k)) such that Y carries large mutual
information with each of the features F(x_i).  These seem different
but related.  The first seems more "on line" while the second seems
more "batch" but both seem to be getting at the same thing, especially
when Y is low dimensional.

Another comment about top level spin involves the Donsker-Varadhan
representation of KL divergence (equation (2) in the paper).  The
paper states that this is not used in the experiments.  This suggests
that it was tried and failed.  If so, it would be good to report this.
Another contribution of the paper seems to be that the mutual
information estimators (4) and (5) dominate (2) in practice.  This
seems important.

---

> ### Author Response · Authors · 2018-11-22
> **Thank you for your thorough review.**
>
> We’re delighted that this approach excites you, and hopefully the comments above and revision address your previous and latest concerns.
>
> - On baselines: See "On architectures and baselines" and "Comparisons to CPC above".
> - Overall spin: We never meant to introduce the prior as a means of addressing trivial solutions to the first bullet point. Rather, the prior term is meant to impose constraints on the marginal distribution of the representation. Disentanglement, for example, is an important property in many fields (neuroscience or RL for instance), and prior matching is a common method for this (e.g., ICA).
> - Ablation studies: See Figure 10, last subfigure in the revision for the ablation study you requested. The prior term has only a small effect on classification accuracy, yet has a strong effect on dependence (it decreases it), according to the NDM measure. If you feel this should be included in the main text, we can add it before the final revision deadline.
> - On the role of the global term: it is true that alone the global term can exhibit some degenerate behavior, and this is especially apparent by classification results. However, its use depends what the end-goal of the representation is. For example, a combined global local version of DIM improves both reconstruction and mutual information estimates considerably over one or the other (Table 4 in the revision). We feel that the global term can still be useful, but it does seem like the global objective without the local objective is not useful.
> - On the DV representation: our initial experiments showed very poor DV performance, but this changed recently when we adopted the strategy of using a very large number of negative samples as in NCE. However, this approach performs only comparably or worse than using the JSD (Tables 1 and 2 in the revision), supporting our claim that the JSD is better for this task. In addition, we added DV to Figure 9, which shows that DV performance decays quickly as fewer images are used in negative sampling.

---

> ### Author Response · Authors · 2018-11-28
> **On SOTA comparisons**
>
> Thank you for your updated review. We actually had an internal debate about how to best phrase this, as we don't want to overclaim anything. Your suggested edit is better, and we will change the sentence at the next revision opportunity.

---

### Official Review · AnonReviewer3 · 2018-11-04
**Mutual information-based representation learning with additional tricks for performance gain.**

**Rating:** 7
**Confidence:** 4

**Review:**

This paper presents a representation learning approach based on the mutual information maximization.
The authors propose the use of local structures and distribution matching for better acquisition of representations (especially) for images.

Strong points of the paper are:
* This gives a principled design of the objective function based on the mutual information between the input data point and output representation.
* The performance is gained by incorporating local structures and matching of representation distribution to a certain target (called a prior).

A weak point I found was:
The local structure and evaluation are specialized for classification task of images.

Questions and comments.
* Local mutual information in (6) may trivially be maximized if the summarizer f (E(x) = f \circ C(x) with \psi omitted for brevity) concatenates all local features into the global one.
How was f implemented? Did you compare this concatenation approach?
* Can we add DIM like a regularizer to an objective of downstream task?
It would be very useful if combining an objective of classification/regression or reinforcement learning with the proposed (8) is able to improve the performance of the given task.
* C^(i)_\psi(X) in (6), but X^(i) in (8): are they the same thing?

---

> ### Author Response · Authors · 2018-11-22
> **Thank you for your review:**
>
> Key points:
> - Image only: As the structural assumptions are important to the MI maximization task of DIM, we wanted to do an in-depth analysis and comparison in this setting. The core ideas of DIM transfer very easily, however, and we anticipate these ideas being successful in the NLP, graph, and RL settings, for example.
>
> Minor comments:
> - Trial solutions: This is true (see discussion with Reviewer 1) and obviously we need a bottleneck or noise in the global variable. One potential solution to this is presented in our occlusion experiments (Table 5 in the revision), where some local features are masked out from computation of the global objective.
> - Using DIM with supervised learning: It sounds reasonable to use DIM directly as a regularizer for supervised learning, and our fine-tuning experiments for STL10 support this. However, we have not tried this experiment specifically.
> - C and X: C_i is the feature map location that corresponds to the receptive field X_i.

---

### Official Review · AnonReviewer1 · 2018-11-05
**Interesting take on representation learning, but text needs improvement**

**Rating:** 7
**Confidence:** 5

**Review:**

This paper proposes Deep InfoMax (DIM), for learning representations by maximizing the mutual information between the input and a deep representation. By structuring the network and objectives to encode input locality or priors on the representation, DIM learns features that are useful for downstream tasks without relying on reconstruction or a generative model. DIM is evaluated on a number of standard image datasets and shown to learn features that outperform prior approaches based on autoencoders at classification.

Representation learning without generative models is an interesting research direction, and this paper represents a nice contribution toward this goal. The experiments demonstrate wins over some autoencoder baselines, but the reported numbers are far worse than old unsupervised feature learning results on e.g. CIFAR-10. There are also a few technical inaccuracies and an insufficient discussion of prior work (CPC). I don't think this paper should be accepted in its current state, but could be persuaded if the authors address my concerns.

Strengths:
+ Interesting new objectives for representation learning based on increasing the JS divergence between joint and product distributions
+ Good set of ablation experiments looking at local vs global approach and layer-dependence of classification accuracy
+ Large set of experiments on image datasets with different evaluation metrics for comparing representations

Weaknesses:
- No comparison to autoencoding approaches that explicitly maximize information in the latent variable, e.g. InfoVAE, beta-VAE with small beta, an autoencoeder with no regularization, invertible models like real NVP that throws out no information. Additionally, the results on CIFAR-10 are worse than a carefully tuned single-layer feature extractor (k-means is 75%+, see Coates et al., 2011).
- Based off Table 9, it looks like DIM is very sensitive to hyperparameters like gamma for classification. Please discuss how you selected hyperparameters and whether you performed a similar scale sweep for your baselines.
- The comparison with and discussion of CPC is lacking. CPC outperforms JSD in almost all settings, and CPC also proposed a "local" approach to information maximization. I do not agree with renaming CPC to NCE and calling it DIM(L) (NCE) as the CPC and NCE loss are not the same. Please elaborate on the similarties and differences!
- The clarity of the text could be improved, with more space in the main text devoted to analyzing the results. Right now the paper has an overwhelming number of experiments that don't fit concisely together (e.g. an entirely new generative model experimentsin the appendix).

Minor comments:
- As noted by a commenter, it is known that MI maximization without constraints is insufficient for learning good representations. Please cite and discuss.
- Define local/global earlier in the paper (intro?). I found it hard to follow the first time.
- Why can't SOMs represent complex relationships?
- "models with reconstruction-type objectives provide some guarantees on the amount of information encoded": what do you mean by this? VAEs have issues with posterior collapse where the latents are ignored, but they have a reconstruction term in the objective.
- "JS should behave similarly as the DV-based objective" - do you have any evidence (empirical or theoretical) to back up this statement? As you're maximizing JSD and not KL, it's not clear that DIM can be thought of as maximizing MI.
- Have you tried stochastic encoders? This would make matching to a prior much easier and prevent the introduciton of another discriminator.
- I'm surprised NDM is much smaller than MINE given that your encoder is deterministic and thus shouldn't throw out any information. Do you have an explanation for this gap?
- there's a trivial solution to local DIM where the global feature can directly memorize everything about the local features as the global feature depends on *all* local features, including the one you're trying to maximize information with. Have you considered masking each individual local feature before computing the global feature to avoid this trivial solution?

-----------------------

Update: Apologies for the slow response. The new version with more baselines, comparisons to CPC, discussion of NCE, and comparisons between JS and MI greatly improve the paper! I've increased my score (5 -> 7) to reflect the improved clarity and experiments.

---

> ### Author Response · Authors · 2018-11-10
> **Similarities and differences between DIM and CPC and the use of the term "NCE"**
>
> We will provide a complete rebuttal soon, but first we address some concerns about our use of the terms DIM/CPC/NCE etc.
>
> DIM(L) and CPC have many similarities, but they are not the same. The key difference between CPC and DIM is the strict way in which CPC structures its predictions, as illustrated in Figure 1 of [1]. CPC processes local features sequentially (fixed-order autoregressive style) to build a partial “summary feature”, then makes separate predictions about several specific local features that weren’t included in the summary feature.
>
> For DIM (without occlusions), the summary feature is a function of all local features, and this “global” feature predicts all of those features simultaneously in a single step, rather than forming separate predictions for a few specific features as in CPC. A consequence of this difference is that DIM is more easily able to perform prediction across all local inputs, as the predictor feature (global) is allowed to be a function of the predicted features (local). DIM with occlusions shares more similarities with CPC, as it mixes self-prediction for the observed local features with orderless autoregression for the occluded local features (see [6] for further discussion of ordered vs orderless autoregression).
>
> Using Noise Contrastive Estimation (NCE) to estimate and maximize mutual information was first proposed in [1], and we credit them in the manuscript (and we will further emphasize this in the revision). While there are a variety of NCE-based losses [2, 3, 4], they all revolve around training a classifier to distinguish between samples from the intractable target distribution and a proposal noise distribution. E.g., [5] uses NCE based on an unbalanced binary classification task, and the loss in [1] is a direct extension of this approach. While novel to [1], we do not consider this NCE-based loss the defining characteristic of CPC, which could instead use, e.g. the DV-based estimator proposed in [7].  The authors of [1] specifically mention this as a reasonable alternative. Due to significant differences in which mutual informations they choose to estimate and maximize, we think it would be ungenerous to consider our method equivalent to CPC whenever we use this estimator.
>
> [1] Oord, Aaron van den, Yazhe Li, and Oriol Vinyals. "Representation learning with contrastive predictive coding." arXiv preprint arXiv:1807.03748 (2018).
> [2] Gutmann, Michael, and Aapo Hyvärinen. "Noise-contrastive estimation: A new estimation principle for unnormalized statistical models." Proceedings of the Thirteenth International Conference on Artificial Intelligence and Statistics. 2010
> [3] Gutmann, Michael U., and Aapo Hyvärinen. "Noise-contrastive estimation of unnormalized statistical models, with applications to natural image statistics." Journal of Machine Learning Research 13.Feb (2012): 307-361.
> [4] Mnih, Andriy, and Yee Whye Teh. "A fast and simple algorithm for training neural probabilistic language models." arXiv preprint arXiv:1206.6426 (2012).
> [5] Mikolov, Tomas, et al. "Distributed representations of words and phrases and their compositionality." Advances in neural information processing systems. 2013.
> [6] Benigno Uria, Marc-Alexandre Cote, Karol Gregor, Iain Murray, and Hugo Larochelle. “Neural Autoregressive Distribution Estimation.” arXiv preprint arXiv:1605.02226 (2016).
> [7] Mohamed Ishmael Belghazi, Aristide Baratin, Sai Rajeshwar, Sherjil Ozair, Yoshua Bengio, Aaron Courville, Devon Hjelm ;Proceedings of the 35th International Conference on Machine Learning, PMLR 80:531-540, 2018.

---

> ### Author Response · Authors · 2018-11-15
> **Followup on the loss function we and CPC call "NCE"**
>
> While searching for more prior work based on different versions of the original "binary" form of NCE, we found an explicit presentation of the "multinomial" NCE used in CPC and DIM.
>
> The loss presented in CPC is less novel than we previously thought. The multinomial version of NCE is precisely described in Section 3 of [1]. A rigorous analysis of the relation between binary and multinomial NCE was also recently published in [2, page 3], which was submitted for review prior to CPC's appearance on arXiv.
>
> [1] "Exploring the Limits of Language Modeling" (Jozefcowicz et al., 2016)
> [2]  "Noise Contrastive Estimation and Negative Sampling for Conditional Models: Consistency and Statistical Efficiency" (Ma and Collins, EMNLP 2018),

---

> ### Author Response · Authors · 2018-11-22
> **Official rebuttal**
>
> Thank you for your detailed review, and we hope that our revisions address your concerns.
>
> Key points:
> - No comparison to autoencoders/beta-VAE/etc: See our discussions above under “New baselines”. We’ve now added these comparisons for classification results.
> - DIM vs CPC: see our previous comments on differences between CPC and DIM, as well as usage of the softmax-type “NCE”.
> - Comparison to CPC: See discussion above “Comparisons to CPC”.
> - Weak performance compared to older methods on Cifar10 (e.g. Coates et al., 2011): See discussions above “On architectures and baselines”. We have added some more details w.r.t. other models in Section 4.2, in “classification comparisons”. Also see "Comparisons to CPC" for improved results on CIFAR10.
> - NCE versus JSD: It would be difficult to conclude that NCE is uniformly superior. While NCE tends to be superior with a large number of negative samples, the differences diminish with larger datasets (Table 2). In addition, JSD outperforms NCE as you reduce the number of images used as negative samples (Figure 9). This will play a factor when choosing the right loss, as more negative samples means more computations / more memory in order to compute the softmax.
> - Sensitivity of the beta term: There was an error in the ranges presented in Figures 9 (accidentally cropped). The last subfigure shows that there is relative insensitivity of gamma (prior term), and much more sensitivity to beta (local term). The performance variation is only ~1% across the gamma, which is not enough to change conclusions of baseline comparisons.
> - We modified the text to improve clarity w.r.t. comments from all reviewers. Many of the experiments we put into the Appendix were related to questions we had about the model / representation, and we excluded them from the main text precisely because they do not relate directly to the main story. However, we chose to keep them in the Appendix as we found them interesting and informative.
>
> Minor comments:
> - On mutual information and constraints: See the updated version of Section 2, next to last paragraph.
> - On local definition: We have modified the text in the first paragraph to help define the “local” MI objective earlier.
> - On SOM: We modified this sentence to read "generally lack the representational capacity of deep neural networks".
> - On reconstruction and MI / VAEs: There was an error in Equation (1), which has now been fixed.
> - On the JSD and the mutual information: This is an important point, and we added a discussion appendix A1 to show the JSD between the joint and the product of marginals is related to PMI as well as some empirical analysis under a discrete setting.
> - Stochasticity: We have tried dropout as a form of stochasticity, and this does not significantly change classification performance, though it is reasonable to posit this might affect the encoder’s ability to match the marginal output to a given prior.
> - NDM vs MINE: NDM is small as the prior term is adversarial and is encouraging the aggregated posterior to match the prior. Small NDM indicates more independence / disentanglement, which is the desired effect (see Figure 12 for the study with beta VAE). DIM encourages the MINE measure to be large, though a combined global / local objective works best (Table 3). There is no straightforward direct relationship between disentanglement and mutual information.
> - Trivial solutions: Trivial solutions are a possibility, and surely this risk increases as the size of the global vector increases, though we never ran across this issue in our experiments (the dimension of 64 was chosen somewhat arbitrarily and to match other latent space sizes, such as those found in GANs. Our limited experiments with larger global vectors had no issues). The experiment you describe is nearly identical to our occlusion experiments (Table 5), which do indeed improve classification performance. It is reasonable to posit that other occlusion-type tasks would modify the representation in desirable ways.

---

### Public Comment · (anonymous) · 2018-10-13
**Missing reference on InfoMax-based unsupervised learning**

It has been already pointed out that InfoMax alone is not enough to learn useful representations [1][2]. [1][2] apply regularization to resolve this problem, and your method can be also regarded as (a different kind of) regularization.

[1] Gomes, R., Krause, A., and Perona, P. Discriminative clustering by regularized information maximization. In NIPS, 2010.
[2] Hu, W., Miyato, T., Tokui, S., Matsumoto, E., and Sugiyama, M. Learning discrete representations via information maximizing self-augmented training. In ICML, 2017.

---

> ### Author Response · Authors · 2018-10-15
> **DIM and regularization**
>
> Thank you for the references. The works cited essential do the global version of DIM, but with discrete representations rather than continuous. Solutions for "global" infomax become degenerate, which motivates the use of regularization in the encoder. Using regularization such as those used in the referenced works (weight decay in [1] and data augmentation [2]) is essential for these approaches to work. This problem also affects us, and this probably is the reason for poor performance of "global DIM" with deterministic input->representation mappings.
>
> We find that the regularization used in [2] is far more relevant to our work, as it "regularizes" the model by making it more robust to data augmentation / sensible transformation at the input space. This is similar in spirit to what we do in the occlusion experiments, where augmentation is done by removing part of the input when computing the global vector.  Overall, [2] is essentially equivalent to adding data augmentation to the global version of DIM in the discrete setting. While the goal of the local version of DIM is to improve generalization by spatial consistency across features, the connection to data augmentation in [2] is not as clear-cut. We do agree that [2] is highly relatable to our work and will add it in the related works on the topic of "leveraging known structure" / data augmentation.

---

### Author Response · Authors · 2018-11-22
**Thank you for your helpful feedback**

We thank the reviewers for providing productive comments and critiques. We believe this input has improved the quality of our work. We first address key shared concerns, and then respond to specific points from individual reviews.

On architectures and baselines:
Our baselines and architectures were chosen to provide a level comparison across methods, rather than to maximize performance of our method. We tried to stay true to common / popular architectures from papers on unsupervised representation learning -- namely, DCGAN- and Alexnet-type encoders. We did not perform significant hyper-optimization on these architectures. For the classification results, our method and all baselines were trained in the same setting with the same architecture. The CIFAR10 supervised results are poor compared to SOTA results that rely on data augmentation and more sophisticated architectures. We did not intend to mislead. We modified Section 4.2 to help readers correctly interpret comparisons with supervised results. To our knowledge, our STL-10 results are SOTA for the unsupervised setting.

New baselines:
We have included new baselines to address concerns from Reviewer 1: CPC, beta VAE with low beta, and an unregularized autoencoder. See Tables 1, 2, and 3 in the revision. We did not implement NICE or real NVP as these involve specialized architectures. Following the same settings as our existing baselines, DIM(L) significantly outperformed all new baselines in classification results. The overall effect of beta in beta VAE is unremarkable. We report results for beta=0.5, which performed best, but also tested beta in {0.01, 0.1, 0.2, 0.5}.

Comparisons to CPC:
We spent considerable time implementing a CPC baseline, and had difficulty getting results that were significantly better than even BiGAN in our test setting. To achieve strong results with CPC, we needed to use an encoder architecture closer to that in the CPC paper. Specifically, we extract each local feature from a patch cropped from the full image. The patches form a 7x7 grid and have 50% overlap between neighboring patches. With this architecture, DIM(L) outperforms CPC on CIFAR10 using a ResNet-type encoder for the cropped patches. When classifying based on the full 7x7 grid of local features, DIM(L) achieves 80.9% accuracy and CPC achieves 77.5%. When strided crops were used with data augmentation on STL10, DIM(L) and CPC performed comparably, both achieving ~77% without fine-tuning through the Alexnet encoder. When we used a version of DIM with multiple global representations using a single convolutional layer, DIM got over 78%. Some of these differences could be architectural so DIM and CPC are at worst comparable in this setting, but we can conclude that the complex strictly ordered autoression in CPC is unnecessary. We have added a paragraph to Section 4.2, in “classification comparisons” to discuss these comparisons.

---

### Meta-Review · Area_Chair1 · 2018-12-13

**Confidence:** 4
**Recommendation:** Accept (Oral)

**Metareview:**

This paper proposes a new unsupervised learning approach based on maximizing the mutual information between the input and the representation. The results are strong across several image datasets. Essentially all of the reviewer's concerns were directly addressed in revisions of the paper, including additional experiments. The only weakness is that only image datasets were experimented with; however, the image-based experiments and comparisons are extensive. The reviewers and I all agree that the paper should be accepted, and I think it should be considered for an oral presentation.